# Partner Modelling Emerges in Recurrent Agents (But Only When It Matters)

**Ruaridh Mon-Williams**[1]* **Max Taylor-Davies**[1]* **Elizabeth Mieczkowski**[2] **Natalia Vélez**[2]
**Neil R. Bramley**[1] **Yanwei Wang**[3]† **Thomas L. Griffiths**[2]† **Christopher G. Lucas**[1]†
[1]University of Edinburgh   [2]Princeton University   [3]Massachusetts Institute of Technology

## Abstract

Humans are remarkably adept at collaboration, able to infer the strengths and weaknesses of new partners in order to work successfully towards shared goals. To build AI systems with this capability, we must first understand its building blocks: does such flexibility require explicit, dedicated mechanisms for modelling others—or can it emerge spontaneously from the pressures of open-ended cooperative interaction? To investigate this question, we train simple model-free RNN agents to collaborate with a population of diverse partners. Using the 'Overcooked-AI' environment, we collect data from thousands of collaborative teams, and analyse agents' internal hidden states. Despite a lack of additional architectural features, inductive biases, or auxiliary objectives, the agents nevertheless develop structured internal representations of their partners' task abilities, enabling rapid adaptation and generalisation to novel collaborators. We investigated these internal models through probing techniques, and large-scale behavioural analysis. Notably, we find that structured partner modelling emerges when agents can influence partner behaviour by controlling task allocation. Our results show that partner modelling can arise spontaneously in model-free agents—but only under environmental conditions that impose the right kind of social pressure.

## 1   Introduction

While humans are certainly impressive 'solo' learners and problem-solvers, our capacity for cooperation and collaboration is even more remarkable—enabling us to achieve goals beyond the reach of any single individual, and leverage the complementary abilities of others while sharing or mitigating the costs of action. In particular, humans display exceptional *flexibility* in adapting to unfamiliar partners and task contexts. Indeed, it could be argued that this capacity for flexible collaboration is one of the primary contributors to the success of our species—without it, it is hard to see how our ancestors could have developed the culture and civilisation that persist to this day [1, 2, 3]. As we develop artificial agents that will operate alongside us, occupying our homes and workplaces, it is crucial that they too can share in this collaborative process [4].

One explanation for the powerful flexibility of human collaboration lies in our well-developed 'Theory of Mind' (ToM)—our general faculty for inferring and representing the latent mental properties (such as goals, beliefs, desires or intentions) that drive others' behaviour [5]. As with many other social contexts, effective collaboration with diverse partners requires an individual to not only react to the actions of others, but also attempt to *predict*, and where appropriate to *influence* them—processes which rely on the formation of predictive representations ('mental models') of other agents' decision-making [6, 7, 8]. In collaborative contexts, we can use our ToM to infer and represent the varying

---

*Equal contribution. Correspondence to `ruaridh.mw@ed.ac.uk` or `m.taylor-davies@sms.ed.ac.uk`
†Equal senior authorship

39th Conference on Neural Information Processing Systems (NeurIPS 2025).

strengths and weaknesses of different partners [9, 10, 11]. For example, imagine being assigned to work with unfamiliar classmates on a group project for a machine learning class. Effectively dividing up the different tasks will require representations of each contributor's abilities: who will be best suited to implement the code, write the report, and deliver the presentation.

In this paper, we examine whether artificial agents, when trained to collaborate with different partners but without explicit mechanisms for agent modelling, spontaneously develop internal representations of their partners' abilities. We investigate this question in a fully cooperative setting, where agents optimise a shared goal (i.e., a single reward function), but have no prior knowledge of each other's attributes or action policies. Crucially, we train reinforcement learning agents with generic recurrent architectures and only task reward supervision—there are no auxiliary objectives or architectural priors pushing agents to model one another. This stands in contrast to prior work, such as Rabinowitz et al.'s 'Machine Theory of Mind' framework, which relies on specialised components optimised explicitly to infer other agents' internal states [12]. We find that despite these minimal inductive biases, agents develop structured, internal representations that **(i)** encode the different competencies of their partners; **(ii)** generalise to previously unseen collaborators; and **(iii)** emerge selectively, depending on agents' ability to control task allocation. Together, our findings suggest that partner modelling can arise within artificial agents solely from the demands of flexible cooperation, *without* explicit incentives or specialised architectures.

## 2 Related work

### 2.1 Ad hoc teamwork

The field of ad hoc teamwork (AHT) deals with the problem of developing agents that learn to collaborate 'on the fly' with previously unseen 'teammates', without any prior coordination [13]. AHT shares some basic elements with the field of multi-agent reinforcement learning (MARL); but where MARL typically assumes control of all agents in the environment, in AHT we control only a single agent (often called the 'AHT agent' or 'learner'; we will use 'ego agent' throughout), with teammates' actions governed by either simple heuristics or pre-trained (frozen) RL policies. AHT also considers only settings where all agents share a common cooperative objective—while individual agents might have additional goals or small differences in reward function, they are never in conflict with one another. The focus of research in AHT has mainly been on producing agents that can adapt to the varying 'play styles' (policies) of different partners or human collaborators [14]. In contrast to self-play training (where agents co-adapt to each other), ad hoc agents must generalise to novel partners under zero shot (no prior interaction) or few shot (minimal adaptation rounds) conditions. AHT can thus be viewed in large part as the problem of rapidly inferring a novel teammate's underlying parameters or characteristics. Accordingly, many approaches have involved explicit inference and representation of these characteristics; traditionally via forms of Bayesian belief-updating over a discrete teammate space [15, 16, 17, 18], or more recently using neural-network-based encoders to learn latent representations of teammate policies [12, 19, 20]. In contrast, our work uses the AHT setting to study *implicit*, emergent partner modelling in simple RNN agents without additional architectural components or auxiliary objectives.

### 2.2 Agent modelling

While many AHT approaches leverage some method for representing different teammates, the problem of modelling other agents in a shared environment is not unqiue to the AHT setting [21]. Recent work on agent modelling has primarily employed deep neural network-based approaches, where a dedicated module is optimised explicitly to produce useful representations of other agents' properties via some auxiliary objective. These representations can then be used to condition the controlled agent's own action policy [21, 22], allowing them to adapt their behaviour directly to the properties of the different agents with whom they must interact. For example, He et al. [23] extend the DQN architecture with an additional network that produces representations of the opponent policy. In contrast, Raileanu et al. [24] avoid having to maintain a separate model of other agents by using the learner's own current policy to infer others' goals via maximum likelihood. Numerous other works have employed some form of encoder-decoder architecture, typically trained via a reconstruction loss to learn latent embeddings that facilitate behaviour prediction [25, 12, 19, 26, 27, 28].

Beyond these explicit modelling approaches, a parallel line of work has examined social influence and coordination in multi-agent RL. For example, Jaques et al. [29] show that giving agents intrinsic motivation to shape others' behaviour improves cooperative outcomes, while work on zero-shot coordination demonstrates that agents trained with diverse partners can adapt to unseen teammates without additional training [30, 31]. These studies highlight the general idea that social prediction and adaptation are key to successful collaboration.

## 2.3 Emergent representations in model-free RL

In contrast to the explicit approach common across most agent modelling research, a different line of work has explored the *implicit* representations that emerge spontaneously in model-free RL agents trained only to achieve a particular high-level task. For example, multiple works have investigated the representations that develop within RL agents trained on simple navigation tasks, finding for example that agents encode target distance, reachability, and progress from their starting location [32], and that structured 'mental maps' of the environment emerge in the memories of 'blind' RNN agents [33]. Other research has explored the information encoded by model-free puzzle-solving or game-playing agents, isolating goal representations in maze-solving networks [34], humanlike chess concepts in AlphaZero [35], and planning-like abilities in RNN agents trained to play sokoban [36].

Our work is also closely connected to the meta-learning literature. Recurrent policies trained across many tasks or partners can act as implicit meta-learners, encoding past experience in hidden states to support rapid adaptation [37]. This view frames recurrence as a way for agents to learn about collaborators, not just task features. Our findings extend this perspective by showing that such implicit meta-learning can give rise to partner-specific internal models under the right collaborative pressures.

## 2.4 Cognitive and economic perspectives

Work in cognitive science, anthropology, and economics converges on the idea that prediction underpins intelligence and collaboration. Clark [38] argues that perception and action are driven by hierarchical prediction. Byrne and Whiten's 'Machiavellian Intelligence' hypothesis [? ] proposes that human intelligence evolved to anticipate others' behaviour. Harsanyi [39] formalised this idea in economics through Bayesian games, where agents reason about hidden partner traits. Grosz and Kraus [40] further emphasise the need for shared predictive structures to coordinate group plans. Our results align with these perspectives, showing that predictive partner models can arise spontaneously in simple recurrent agents under collaborative pressure.

## 2.5 Attention

Recent work uses attention-based architectures to study social reasoning. Long et al. [41] analyse collaboration via attention weights, while Decision Transformer [42] frames RL as sequence modelling. In contrast, we use a simple RNN to show that structured partner representations can emerge without strong inductive biases. Future work could apply attention mechanisms to probe whether agents implicitly track partner positions or trajectories.

# 3 Problem Formulation

As is standard in the AHT literature, we formulate the problem as a two-agent partially observable Markov decision process (POMDP). This is defined by the tuple $\mathcal{M} = \langle \mathcal{S}, \mathcal{O}_1, \mathcal{O}_2, \mathcal{A}_1, \mathcal{A}_2, P, r, \gamma \rangle$, where $\mathcal{O}_i$ and $\mathcal{A}_i$ denote the observation and action spaces of agent $i \in \{1, 2\}$ (with $\vec{\mathcal{O}} = \mathcal{O}_1 \times \mathcal{O}_2$, $\vec{\mathcal{A}} = \mathcal{A}_1 \times \mathcal{A}_2$), $\mathcal{S}$ is the environment state space, $P : \mathcal{S} \times \vec{\mathcal{A}} \mapsto \Delta(\mathcal{S})$ denotes the state transition function, $r : \mathcal{S} \times \vec{\mathcal{A}} \mapsto \mathbb{R}$ is the shared reward function, and $\gamma$ is the discount factor. At each timestep $t$ the ego agent (agent 1) receives an snapshot of the environment $o_t^1 \in \mathcal{O}_1$—information from which may be retained in future timesteps as part of internal memory state $h_t$ with a recurrent function $h_t = f(h_{t-1}, o_t^1)$. The ego agent acts according to its learned policy $\pi(a_t^1 \mid h_t)$ and the partner (agent 2) acts according to its (fixed) pre-trained policy $\pi_2$, governed by latent parameters $z^* \in \mathcal{Z}$, sampled from distribution $p(z^*)$ at the beginning of each episode. These traits, such as how quickly a given partner can perform each task, are not directly observable to the ego agent—rather, as in

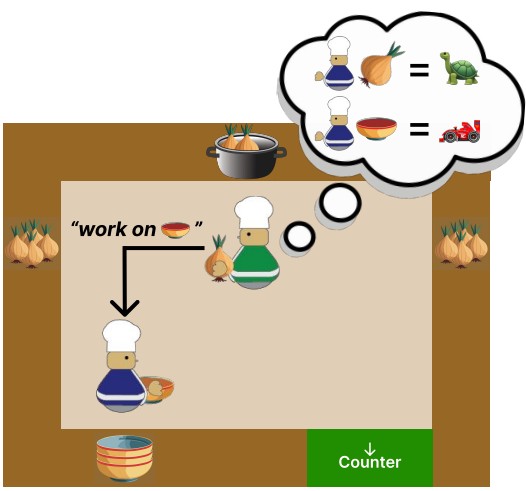

Figure 1: An illustration of our task setting, based on the 'cramped room' layout of Overcooked-AI. The ego agent (green) has a learned internal representation of how competent (fast) their partner (blue) is at each of the two subtasks; it uses this representation to determine which subtask the partner should work on.

the context of human theory of mind, they must be inferred (explicitly or implicitly) from observed behaviour.

The ego agent is trained *only* to maximise the expected cumulative reward across episodes, $\max_\pi \; \mathbb{E}_{z^* \sim p(z^*)} \; \mathbb{E}_{\tau \sim \pi, \pi_2(z^*)} \left[ \sum_{t=0}^{T} \gamma^t r_t \right]$, where $\tau$ denotes a trajectory of states, observations, and actions sampled from the ego policy $\pi$ and partner policy $\pi_2(z^*)$ under the transition dynamics $P$. Each episode involves a different partner sampled from a distribution over latent traits $z^* \in \mathcal{Z}$. Importantly, no explicit architectural mechanisms or auxiliary objectives encourage modelling of these latent traits. We are interested in whether, under these minimal conditions, internal partner models nevertheless emerge *implicitly* within the ego agent's recurrent state $h_t$.

## 4 Methods

### 4.1 Environment

Critical to any test of our hypothesis is that the environment imposes *collaborative pressure*; i.e. the ego agent's optimal policy depends on the latent characteristics of their partner, and so modelling those characteristics is conducive to achieving high joint reward. We provide this through Overcooked-AI [43], a fully cooperative environment where agents work together to prepare soups, tasked with maximising the throughput $r = \frac{\Delta \text{Soup}}{\Delta \text{Time}}$ (for additional results in a second cooperative environment, see Appendix A). Each agent must navigate a shared kitchen to gather ingredients, cook them, and serve the completed soups — making success heavily dependent on coordination and division of labour. The environment difficulty can be modified via different recipes, which vary in complexity and number of ingredients, and different kitchen layouts, which introduce specific constraints (e.g. encouraging agents to pass items to perform the task successfully, or forcing agents to navigate around each other in cramped spaces). Figure 1 illustrates one such layout.

### 4.2 Agent Architecture

The ego agent's policy is implemented as a gated recurrent unit (GRU) [44] recurrent neural network (RNN). At each timestep $t$, the agent processes an observation $o_t$ and updates its hidden state via $h_t = GRU(o_t, h_{t-1})$. The hidden state $h_t$ acts as a general dynamic memory, evolving as the interaction unfolds. The ego policy is trained using Proximal Policy Optimisation (PPO) [45], implemented in JAX [46] to facilitate efficient parallel training.

### 4.3 Experimental Design

Achieving high reward in the Overcooked-AI environment requires agents to successfully coordinate two different subtasks: preparing ingredients ('task 1'), and serving soup ('task 2'). We train our ego agent alongside a distribution of partners who vary in how competent they are at the two subtasks (operationalised as how frequently they can take actions towards each task). To introduce a direct connection between partner properties and optimal ego agent behaviour, we grant the ego agent control over which subtask its partner is working towards at any given time. Our hypothesis is that effective task allocation will require the ego agent to learn how to recognise and represent the abilities of different partners—that is, after training, the RNN hidden state dynamics should be optimised to encode how fast a given partner is at tasks 1 and 2 (as illustrated in Figure 1).

Within this framework, we carry out three experiments to probe different dimensions of partner modelling:

#### 4.3.1 Experiment 1: Does the pressure to allocate subtasks drive the emergence of partner modelling?

This experiments tests whether placing the responsibility for subtask allocation to the ego agent encourages it to develop internal representations of its partner's capabilities. In particular, we ask whether the ego agent can learn to reliably allocate tasks effectively by representing different partners' ability profiles? To test this, we generate a distribution of partners, each characterised by a two-dimensional vector $\mathbf{v} = [v_1, v_2]$ denoting the cooldown interval (i.e, the number of timesteps between consecutive actions) for each subtask. These cooldowns can be interpreted as inverse proxies for subtask skill: a lower $v_i$ reflects higher proficiency in subtask $i$, allowing the partner to act more frequently. The ego agent is trained alongside a population of partners with $v_i$ sampled independently from the set $\{1, 2, 3, 4, 7, 9\}$. During evaluation, the ego agent is paired with partners whose cooldowns are sampled from a different set, $\{0, 2, 3, 8, 10\}$. While some individual cooldown values overlap (e.g. 2 and 3), the *pairs* of cooldowns $(v_1, v_2)$ used for training and evaluation are disjoint, ensuring that no test partner configuration was encountered during training. The ego agent has a constant cooldown of 2 for both tasks, allowing for a balance between dominating the tasks (if too fast) and slowing learning (if too slow), and is equipped with an additional action that allows it to dictate which subtask the partner contributes to at each timestep.

#### 4.3.2 Experiment 2: Can agents adapt online to new partners within an episode?

Successful collaboration in the real world often requires us to deal with sudden, unexpected changes. In this experiment, we examine whether our ego agent can adapt dynamically to different partners *within* a single episode. During each training episode, there is a 50% probability that the partner is 'switched' at a random timestep between 30 and 70% of the 600-timestep duration (the random dynamics ensure that the agent cannot memorise a fixed timing pattern). To ensure non-triviality, the post-switch partner is always sampled with a *mirrored* ability profile (i.e. if the initial partner is faster at task 1, the new partner is faster at task 2, and vice versa). When evaluating the agent after training, we simplify the analysis by performing the switch in every episode, always at exactly $t = 300$, and only in a single direction (faster at task 1 $\rightarrow$ faster at task 2).

#### 4.3.3 Experiment 3: Can blind agents develop partner models from task reward alone?

Inspired by the findings of Wijmans et al. [33], who showed that 'blind' agents trained for PointGoal navigation develop internal map-like representations of their environment despite having access only to proprioceptive feedback, we ask a related question—can blind versions of our Overcooked agents, trained purely to maximise cooperative task reward, still develop internal models of their partners' capabilities?

To investigate this, we remove all visual input from our ego agents and restrict them to receiving only egocentric signals (information only about their current grid cell, including their location, orientation, and any objects they are holding). Importantly, they are unable to perceive their partner's location, or directly observe their behaviour.

To generate variation in partner competence, we first train a high-performance onion-preparing agent in self-play, and then inject controlled levels of noise into its policy. This produces a range of partner

behaviours from reliably competent to frequently erratic. The ego agent must infer where on this spectrum each partner lies purely from its own direct experiences and task rewards. Full details of the partner generation and evaluation procedures are provided in the Appendix.

## 4.4 Evaluation Overview

To investigate how and when internal partner models emerge, we analyse agent behaviour across five Overcooked-AI layouts (see Appendix). Our evaluation is motivated by three key questions: (i) Can the ego agent collaborate effectively with previously unseen partners? (ii) Do hidden states encode partner traits such as speed or competence (iii) Does the agent update its internal model in response to mid-episode changes in partner attributes? To address these questions, we analyse overall reward (total number of soups delivered), adaptation curves (how quickly reward accumulates over time), linear probe accuracy (how well partner traits can be 'decoded' from RNN hidden states by optimising a single linear layer with input size $\dim(h)$ and output size 1) and UMAP projections [47] (capturing the structure of hidden states). We also compare against three baselines, each designed to isolate a different factor affecting partner modelling: a *feedforward MLP*, which lacks memory; a *single-partner RNN*, trained on a fixed partner to assess the role of training diversity; and a *non-influential RNN*, trained across the full distribution of partners but without the ability to influence their behaviour. A more detailed description of each baseline is provided in the Appendix.

# 5 Results

## 5.1 Collaborative agents adapt to unseen partners

To establish the importance of modelling different partners to our chosen environment, we compare the performance of our RNN ego agent policies against two simpler baselines: a purely feed-forward MLP policy, and an RNN policy trained alongside only a single constant partner. Figure 2 shows the results of this comparison across five different layouts. We find that the RNN policy trained against diverse partners outperforms both the MLP policy and the single-partner-trained RNN variant; presumably by virtue of a learned ability to model partner parameters. The single exception to this is *fivebyfive_v1*, in which the MLP agent achieved the highest reward—possibly due to the fact that *fivebyfive_v1* is a simpler layout, requiring less spatial coordination and allowing simpler behavioural strategies to perform well. Importantly, the higher performance of the multi-partner-RNN with respect to the single-partner RNN does not reflect simple memorisation of different partners, since the partners encountered in evaluation were unseen during training. In addition to superior performance, we find that evaluation episodes with the (multi-partner-trained) RNN ego agent yield a higher correlation between which task the partner spends most time on, and which task they're fastest at (Figure 2C)—suggesting that the ego agent's task allocation decisions are informed by a representation of different partners' ability profiles.

## 5.2 Agent memory encodes partners' abilities (when there is pressure to do so)

A key idea behind our experimental design is that having the ability to influence which subtask a partner performs will pressure the ego agent to learn hidden representations that encode the task speeds of different partners. To investigate the hypothesis further, we trained RNN policies under three different conditions: our 'full' setup including both partner diversity and task influence, plus the 'single-partner' and 'non-influential' variants described in Section 4.4. For each condition, we then extracted the trained agent's hidden states during evaluation episodes alongisde 46 different unique partners in 5 different layouts (with 20 seeds per condition/layout/partner).

Figure 3A shows, for each condition and layout, a 2D UMAP projection of these hidden states (averaged over the final 50 timesteps of each rollout), coloured by the difference in task speeds for each partner (task 1 speed − task 2 speed). We can see that for the multi-partner-trained RNN, there is a high degree of structure, with less for the single-partner-trained variant, and less still for the non-influential variant. For each condition and layout, we also trained single-layer linear probes at different values of $t$ to predict partner task speeds from hidden states averaged over $(0, t]$. Figure 3B shows the accuracy of these probes on a test set of held-out partners: we see that, for the initial hidden states at $t = 0$, our linear probes perform no better than a baseline trained on random data. As the episodes progress and the ego agent is able to interact with and observe each partner, probe

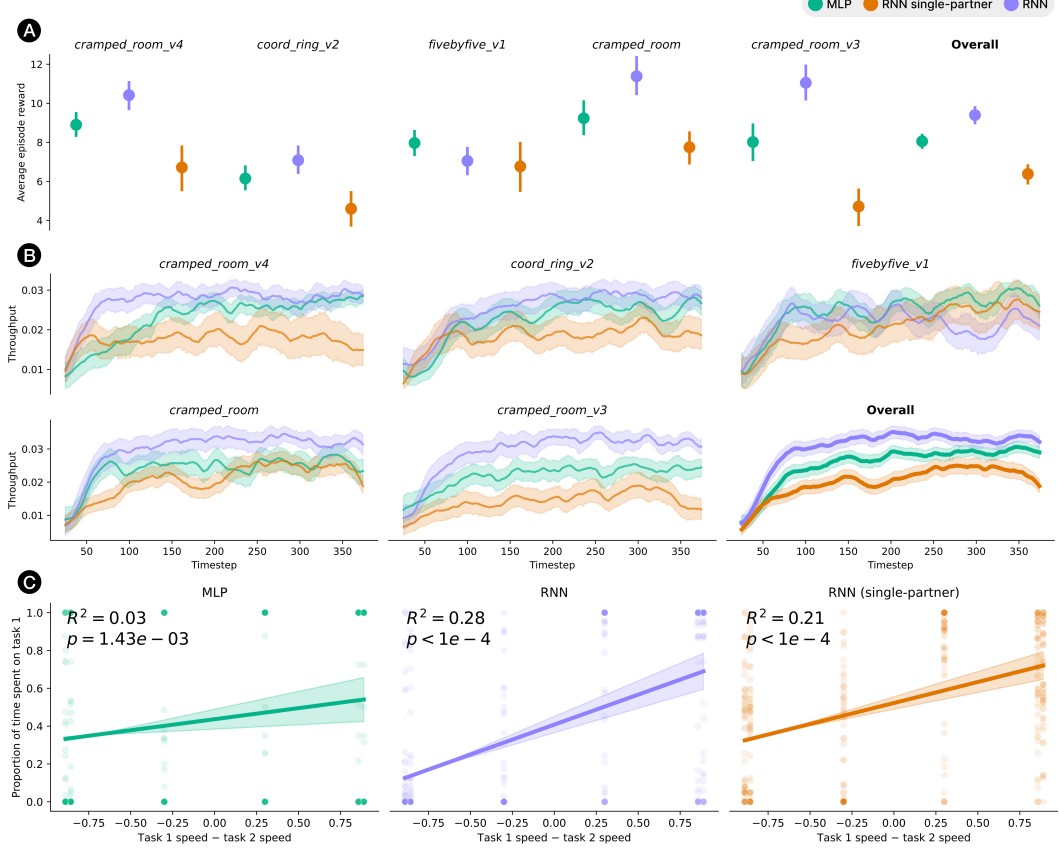

Figure 2: Comparison between different ego agent policies in the overcooked environment. Each policy was evaluated in five different layouts against 8 different combinations of partner speed parameters, over 10 seeds. **(A)** average episode reward, per-layout and overall **(B)** throughput (rate of soup production), per-layout and overall, with shaded areas giving bootstrapped 95% confidence intervals **(C)** correlation between how much faster the partner agent was at task 1 vs task 2 and the proportion of time the partner spent performing task 1 (over all layouts). Shaded areas show 95% confidence intervals over the slope; a higher correlation indicates more efficient task allocation.

accuracy increases across all three conditions—but is highest for the multi-partner-trained RNN, and significantly impaired for the non-influential RNN.

These results offer convincing evidence, first of all, that our ego agent has learned to encode meaningful information about different partners in memory, *without being explicitly trained to do so*. They also demonstrate the importance of environmental pressure to the emergence of these representations. In particular, when the ego agent is stripped of its ability to influence partners' behaviour directly, its hidden states contain significantly less information about partner task abilities (as measured by linear decodeability)—strikingly, even less than those of the ego agent that only ever encountered a single partner during training! For a replication of these results in a second environment, see Appendix A.

### 5.3 Agents can adapt online to new partners

So far, we have established that our RNN ego agents, once trained, can adapt to different partners across different episodes. A stronger version of flexible cooperation involves adapting to new partners 'online', without the environment or internal agent state being reset. To test this, we train an ego agent alongside partners whose task speeds may change up to once per episode. Across three layouts, we then evaluate this agent over a number of 600-timestep rollouts where they are paired initially with a

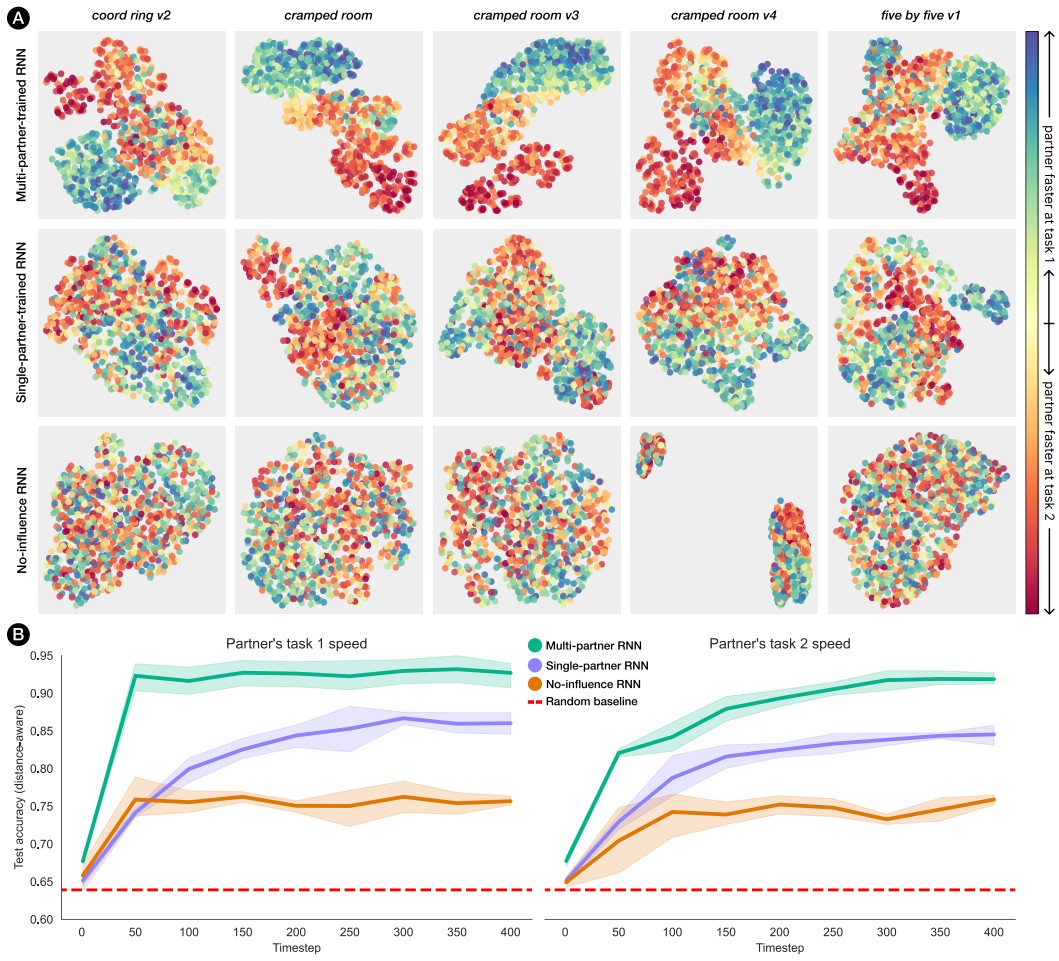

Figure 3: A comparison of the hidden states of RNN ego agents trained under different conditions. **(A)** UMAP embeddings of RNN hidden states averaged over the final 50 timesteps of each episode, coloured by the partner's difference in task speeds (speed 1 − speed 2), for five different layouts of the Overcooked environment. **(B)** Mean test accuracy of linear probes trained to recover partner speeds from sets of RNN hidden states accumulated up to different timesteps (with shaded areas giving bootstraped 95% confidence intervals).

partner that is fast at task 1 and slow at task 2, then switched at $t = 300$ to a partner with the opposite profile.

To measure how well our agent copes with this scenario, we track the average soup throughput over time. From the results in Figure 2B, we expect that the throughput should initially increase to a steady state; we anticipate that it will then drop sharply at $t = 300$ as the partner's task speeds are reversed. After this point, if the ego agent is capable of adapting online to the new partner, the throughput should increase once more to a new steady state; if not, it should remain low. Figure 4A shows that for all three tested layouts, the throughput does indeed increase again after $t = 300$, demonstrating the presence of online adaptation [3]. As further evidence, Figure 4B shows that, on average, the partner is directed to allocate their time mostly to task 1 for $t < 300$ and mostly to task 2 for $t > 300$. Finally, we also visualise in Figure 4C how the ego agent's hidden states are affected by the switch. UMAP projections of hidden states averaged over $t < 300$ align roughly with the distribution of embeddings for 'baseline' episodes with (constant) partners fast at task 1 and slow at task 2; over

---

[3]We note that the throughput does not fully recover post-switch due to an asymmetry in the subtask complexities: because task 1 requires more steps than task 2 to execute, the overall team efficiency is higher when the partner is fast at task 1 and slow at task 2 than the reverse.

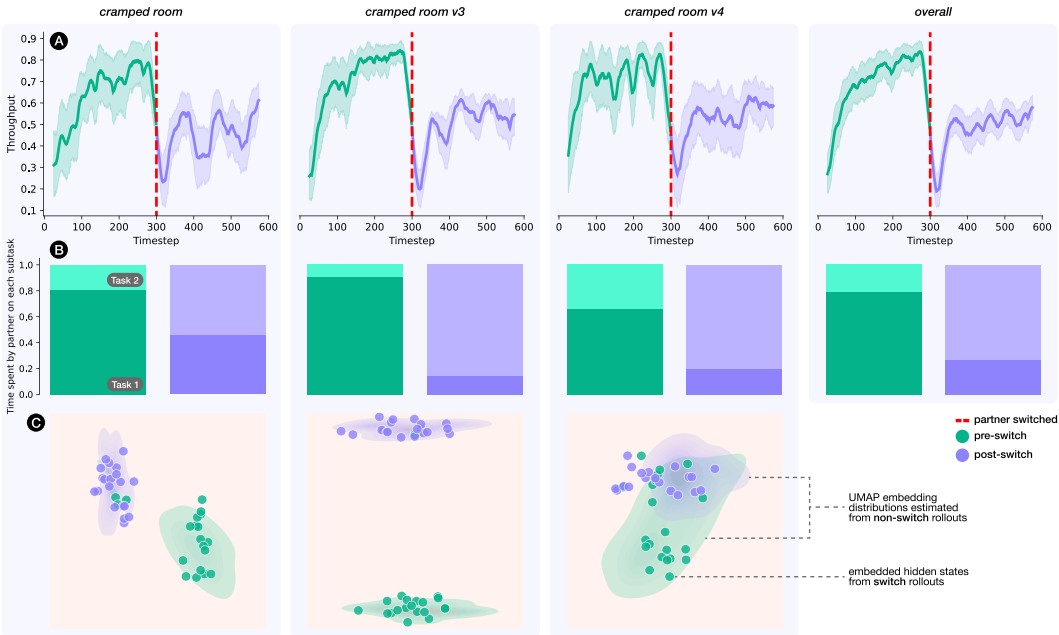

Figure 4: A demonstration of online adaptation. **(A)** Average throughput (rate of soup delivery) during episodes where the partner is switched halfway through from being faster at task 1 to faster at task 2. **(B)** From the same episodes, the average proportion of time spent by the partner performing each subtask before and after the switch. **(C)** UMAP embeddings of the average pre-switch and post-switch RNN hidden states from each episode. Also shown are the distributions (approximated via KDE) for embeddings of hidden states from *non-switch* baseline episodes with partners matching the pre- and post-switch speeds respectively.

$t > 300$, they move to match the distribution for episodes with partners slow at task 1 and fast at task 2 (see supplementary videos for visual examples of this adaptation).

### 5.4 Partner modelling emerges even in blind AI agents

We find that structured partner modelling also emerges in blind agents – trained without any architectural biases toward modelling their partner. Despite relying purely on egocentric observations and scalar task reward, these agents generalise to new partners and outperform both a recurrent agent trained on a single partner and a memoryless MLP baseline. Evaluated across five Overcooked layouts (with six novel partners per layout and 10 random seeds per permutation), the blind RNN agents trained with collaborators with a diverse range of competencies achieved an average throughput of 9.43 soups per episode. This compares to just 5.8 for the single-partner RNN and 1.08 for the MLP. These results show that the interaction structure and memory can enable adaptive behaviour to the capabilities of partner agents – even under harsh observational conditions. Further details, including corresponding videos and a breakdown of the results, are included in the supplementary materials.

## 6 Discussion

In this paper, we have studied the question of whether representations of other agents' relevant attributes can emerge simply as a result of environmental pressure to collaborate effectively with diverse partners. In the absence of dedicated architectural features or auxiliary objectives, we found that RNN agents trained to play a version of the cooperative game 'Overcooked' nevertheless developed structured internal representations of their partners' task abilities. In an additional experiment, we showed that these representations enable agents to adapt online to new partners within the same episode. Finally, we also demonstrated that the development of structured representations is significantly weakened when agents are denied the ability to influence partner behaviour. Taken as a whole, our results serve to illustrate the idea that social intelligence can emerge from specific environmental

pressures acting in concert with general mechanisms for learning and memory, rather than necessarily relying on unique architectures. We believe that this is important to bear in mind as we seek to develop artificial agents that bridge the gap towards humanlike social cognition and behaviour.

It is notable that structured partner modelling also emerged in blind agents trained without visual input, relying solely on egocentric signals and task reward. Despite having no explicit access to their partner's actions or state, these "blind" agents developed internal representations that enabled them to effectively collaborate with partners displaying a diverse range of competencies. This emerged despite minimal inductive biases and limited observational input. This indicates that the structure of the interaction itself is sufficient to drive the emergence of partner-aware policies.

While we feel our work represents a valuable contribution to the study of emergent partner modelling, we highlight various limitations that might serve as starting points for future research. First and foremost, our experiments used only a single cooperative environment (Overcooked-AI)—while we are confident that our results will generalise to other environments and task settings, an obvious target for future work is to confirm this empirically. Of particular interest would be *open-ended* environments that allow us to study how partner representations evolve over time in a more continuous setting; or those with more complex or overlapping subtasks. Relatedly, future work might study whether our findings scale to more than one teammate—we believe that they should, provided that the environment imposes sufficient pressure (i.e. all teammates' behaviour is relevant to task completion). That said, inference will become more demanding with additional agents, and so the ego agent may learn to rely on shortcuts such as modelling the average behaviour of its teammates.

A further limitation is that we have restricted ourselves to studying relatively simple forms of representation. In the real world, people engage in highly complex modelling of their social partners, including through hierarchical representations that deal with how others are perceiving them in turn. It would be interesting to investigate whether these simple, general agent architectures are capable of acquiring such sophisticated capabilities, and under what environmental conditions. Related to this is the fact that our implementation of 'influence' was very strong, essentially taking the form of direct control. Humans typically influence their collaborators in much more nuanced ways—future work might reduce this gap via some form of communication system, where the partner learns to follow (or ignore) high-level instructions from the ego agent.

Aside from these limitations, a further avenue is to explore transformer-based agents, where attention may offer a natural probe of whether agents implicitly track their partners' positions or strategies. Another direction is to test whether internal partner representations can be transferred between agents and tasks via hidden state initialisation. For example, one could evaluate whether seeding an agent's memory with the final hidden state from a previous rollout improves adaptation when encountering the same partner; probing the portability and generality of the learned representations. Comparing 'transplanted' and 'cold-start' agents would provide insight into the extent to which internal partner models support efficient reuse and generalisation. Another possible avenue would involve direct comparisons between the implicit partner modelling approach we study here and various explicit methods: we expect that the latter would perform better in the specific modelling contexts they were trained for, at the cost of reduced flexibility to changes in task environment or partner attribute space.

Finally, while our primary motivation is the desire for artificial agents capable of collaborating flexibly with humans, we believe that a version of our approach might also be used to shed light on the evolutionary differences in collaborative and cooperative behaviour observed across different animal species. To this end, future work might explore using large-scale evolutionary simulations to further study the interplay of environmental pressures and agent architectures; an approach which has recently proven fruitful for investigating other social behaviours such as altruism [48].

# 7   Acknowledgements

This work was supported by the EPSRC CDT in RAS (EP/L016834/1) and the National Defense Science and Engineering Graduate (NDSEG) Fellowship Program awarded to E.M. We thank J. Shah and many others for their invaluable support and expertise.

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

# Appendix

## A    CoinGame results

To ensure that our findings generalise beyond the specific environment of Overcooked, we conducted some preliminary experiments using a modified version of the JaxMARL suite's CoinGame environment [43].

### A.1    Environment

In our version of CoinGame, two agents (the ego agent and one partner) must work together to collect red and blue coins within a small gridworld (see Figure 5A). At any given timestep, the partner is in either 'red mode' (tries only to collect red coins while ignoring blue) or 'blue mode' (vice versa). Different partners are characterised by a two-dimensional 'skill profile' $[s_r, s_b]$, which controls their probability of successfully executing actions when in each mode (the ego agent has $s_r = s_b = 1$). As in our Overcooked experiments, the ego agent can exert influence by switching their partner's mode. The ego agent is rewarded based on the total number of coins collected ($n_{\text{red}} + n_{\text{blue}}$) over an episode; they are thus incentivised to find the most efficient 'division of labour' between themselves and their partner.

### A.2    Experimental procedure

As in our main Experiment 1 (5.1) we trained ego agents for 1e7 timesteps under four conditions:

1. MLP (ego agent uses a simple MLP in place of an RNN)
2. No-influence RNN (ego agent has no control over which coin type the teammate pursues)
3. Single-partner RNN (ego agent only exposed to a single partner type during training)
4. Multi-partner RNN (ego agent paired with multiple partner types during training and has influence)

During training, the single-agent RNN was always paired with a teammate with skill profile $[0.2, 0.8]$; in all other cases partners were sampled uniformly from the set $\big\{[x, 1 - x] \ \forall \ x \in \{0.2, 0.4, 0.6, 0.8\}\big\}$. During evaluation, partners were always sampled uniformly from the set $\big\{[x, 1 - x] \ \forall \ x \in \{0.1, 0.3, 0.5, 0.7, 0.9\}\big\}$. The ego agent thus never encountered during evaluation a partner they had previously seen in training. From these evaluation episodes we recorded the total number of coins collected, the number of timesteps where the teammate was pursuing the 'correct' coin colour (based on their skill profile), and the hidden states of the ego agent RNN (where applicable). As with our experiments in Overcooked, the hidden states were analysed via UMAP projections and linear probe accuracy.

### A.3    Results

Looking at Figure 5B, we see that the multi-partner-RNN ego agent outperforms the single-partner-RNN agent, which in turn outperforms the no-influence-RNN and the MLP agents—replicating the trend we observed in our Overcooked results. Also corroborating our previous results is Figure 6, which shows that hidden states extracted from the multi-partner-RNN are more structured with respect to partner skill profiles than those from the single-partner or no-influence variants. Finally, we computed the correlation to partner skill profile of *individual* RNN hidden unit activations. As Figure 7 shows, we found multiple units with strong positive or negative correlations (16/32 with $|\text{corr}| \geq 0.5$); as well as some units with close-to-zero correlation (likely encoding task/environment information unrelated to partner properties).

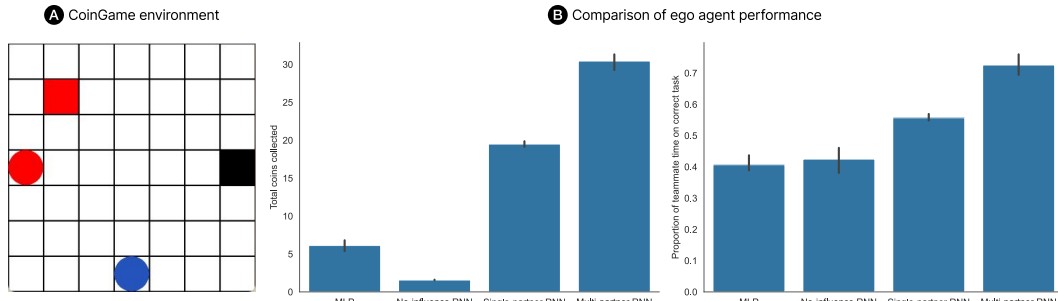

Figure 5: (A) An example CoinGame environment layout at episode start. The two coloured circles are coins, the red square is a teammate currently in 'collect red coins' mode and the black square represents the ego agent. (B) A comparison of the evaluation performance of different ego agents trained in the CoinGame environment, showing the mean total number of coins collected by ego agent and teammate (left) and the mean proportion of time spent by the teammate pursuing the 'correct' coin colour (i.e. the one matched to their highest skill level). Error bars give bootstrapped 95% confidence intervals over 5 random training seeds.

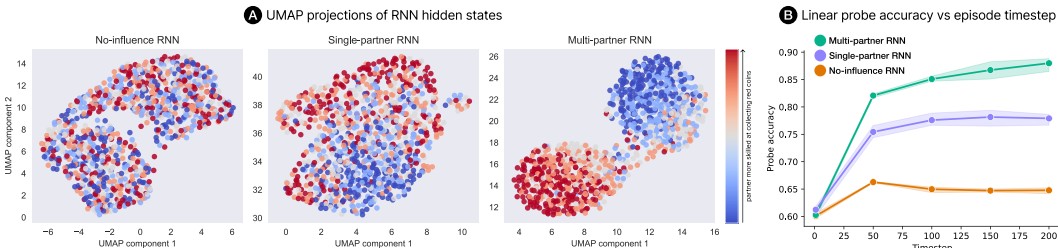

Figure 6: (A) UMAP embeddings of RNN hidden states averaged over the final 50 timesteps of each episode, coloured by partner skill profile. (B) Mean test accuracy of linear probes trained to recover partner skill from sets of RNN hidden states accumulated up to different timesteps (with shaded areas giving bootstraped 95% confidence intervals).

# B   Additional details

## B.1   Training setup

### B.1.1   Ego agent training

The ego policy is trained on a single GPU using Proximal Policy Optimisation (PPO), running synchronously across 256 parallel Overcooked-AI environment instances. Both agent and environment are implemented in JAX with `jax.jit` for accelerated gradient updates and rollout collection. Each rollout lasts 400 timesteps in Experiments 1 and 3, and 600 timesteps in Experiment 2, and agents are rewarded for every successful soup delivery. Training runs for 15 million timesteps against a distribution of partner agents. For the first 5 million timesteps, a decayed reward shaping is used to aid policy learning (rewarding the agent for putting onions in the pot and for cooking the soup when it contains the correct ingredients). For experiments 1 and 2, partner agents could at any given timestep perform one of two subtasks: (1) placing ingredients into a pot, or (2) serving soup (which involved picking up a bowl, ladling the soup, and delivering it). Each subtask is handled by a separate, pre-trained neural network that specialises in the specific behaviour. The RNN ego agent has an additional action that allows it to set the partner's current subtask. For experiment 3 (with the blind agent), the ego policy is trained against a distribution of partners who perform subtask 1 to varying competencies. We randomised the starting state of each episode during both training and testing. The RNN hyperparameters used for the experiments are shown in Table 1.

### B.1.2   Partner agent training

Each partner policy consists of two feed-forward subtask networks trained independently in self-play using PPO with reward shaping. To obtain subtask-specific policies, each network was paired with a

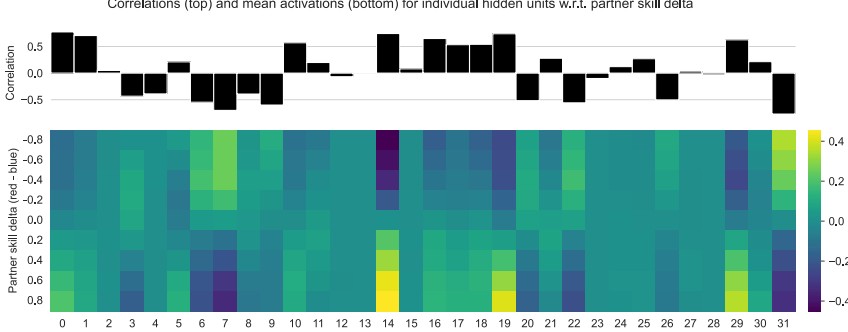

Figure 7: Top: correlation coefficients between individual hidden unit values and partner skill profile computed over 1000 evaluation episodes. Bottom: normalised average value of each hidden unit for each distinct partner.

Table 1: RNN training hyperparameters

| Parameter | Value |
|-----------|-------|
| FC_DIM_SIZE | 128 |
| GRU_HIDDEN_DIM | 128 |
| ACTIVATION | relu |
| LR | 5e-4 |
| ANNEAL_LR | True |
| LR_WARMUP | 0.05 |
| NUM_ENVS | 256 |
| NUM_STEPS | 256 |
| UPDATE_EPOCHS | 4 |
| NUM_MINIBATCHES | 64 |
| TOTAL_TIMESTEPS | 1e7 |
| CLIP_EPS | 0.2 |
| ENT_COEF | 0.01 |
| GAMMA | 0.99 |
| GAE_LAMBDA | 0.95 |
| SCALE_CLIP_EPS | False |
| VF_COEF | 1.0 |
| MAX_GRAD_NORM | 0.25 |

partner that always performed the *other* subtask. To ensure robustness we introduced randomness during training: the subtask network had a 30% probability of waiting on any given step and a 30% probability that its partner would take a random action. This prevented over-fitting to specific interaction patterns or timings and ensured that the learned policies could operate independently. We found that this approach allowed us to produce policies for the partner agent faster than relying on manually specified rules or heuristics.

After training, these subtask networks were combined to form the full partner agents capable of performing both subtasks. To create a distribution of partner behaviours with varying speeds, we controlled the speed at which each subtask policy could be executed when interacting with the ego agent. For example, a partner agent might be able to execute the ingredient placement policy every four timesteps and the serving policy every two timesteps. This enables us to simulate a range of partners with different speeds and proficiency across the two tasks, while utilising the same robust underlying subtask policies.

For experiment 3, we created a distribution of partners with different competencies by adding a certain probability of the partner taking a random action at any given timestep (e.g, a competent partners might have a 5% chance of acting randomly; an incompetent partner might have a 95% chance).

## B.2 Agent architectural design

The architecture includes both input and output fully connected (FC) layers. After the convolutional neural network (CNN) processes the observation, an FC layer maps the resulting embedding to the hidden size of the GRU. The GRU's output is then passed through separate two-layer multilayer perceptrons (MLPs) for the actor and critic heads, which generate action logits and a scalar value estimate, respectively. Observations are pre-processed to ensure each agent has a local, self-centred view of its environment. Furthermore, the CNN output is normalised before being fed into the GRU to stabilise training and improve performance.

## B.3 Evaluation details

### B.3.1 Throughput

The throughput (rate of soup production, used in Figures 2 and 4) is given by $\frac{\Delta \text{Soup}}{\Delta \text{Time}}$. To obtain a point estimate of the throughput at time $t$, we fit the slope of the cumulative reward curve over the sliding window $[t - 25, t + 25]$ by the method of least squares.

### B.3.2 UMAP projections

To obtain 2D embeddings of RNN hidden states (as seen in Figures 3 and 4), we use the official Python implementation of the UMAP algorithm (https://umap-learn.readthedocs.io/en/latest/), with hyperparameters `min_dist=1.0` and `n_neighbors=N-1` where $N$ is the number of hidden state datapoints being projected. These hyperparameters were selected to ensure focus on the 'global', high-level structure of the embedding space wrt the partner parameters, rather than small-scale localised clusterings. For all other hyperparameters we use the library default values.

### B.3.3 Linear probe analysis

For our linear probe analysis (used in Figure 3), we optimise a linear layer to perform the classification task $\vec{x}_t \mapsto y$ where $\vec{x}$ is the averaged RNN hidden states up to time $t$ from a single episode, and $y$ is the partner's speed at *either* task 1 or 2 for that episode. We train probes separately for different values of $t$ as well as ego agents trained under different conditions. Each probe is trained for a total of 1e3 steps using the Adam optimiser with a learning rate of 1e-2. We use a train-test split of 80-20 over rollout seeds (i.e. where we have 20 seeds per partner speed combination, we randomly assign 16 of those seeds to the train set and 4 to the test set). We report the distance-aware classification accuracy over the test set. For comparison, we train a 'baseline' probe under identical conditions using random $\vec{x}$ (sampled from the standard Normal distribution with the same shape as the hidden states) and the true $y$ labels.

## B.4 Other experiment details

### B.4.1 Experiments 1-2

To reduce the chance of poor policy convergence due to a random seed, we ran the entire training process over five random seeds per environment layout, and selected the ego agent policy that achieved the highest average episode return at the end of training. During testing, we paired the ego agent with 24 different partners not encountered during training and collected 20 episode rollouts per partner (over 20 random seeds, which randomised the starting state), for a total of 2400 different rollouts. The seeds that achieved the highest return are as follows:

Table 2: Best performing seed (1–5) for each layout and model

|                  | MLP | RNN | RNN Online | RNN Single Partner |
|------------------|-----|-----|------------|--------------------|
| Cramped Room     | 2   | 3   | 5          | 2                  |
| FiveByFive       | 4   | 2   | 4          | 5                  |
| Coord Ring       | 2   | 3   | 5          | 1                  |
| Cramped Room V3  | 1   | 2   | 2          | 5                  |
| Cramped Room V4  | 2   | 2   | 2          | 5                  |

For experiments 1 and 2, we used an ego agent with a fixed cooldown interval of 2 (across both subtasks). We found that if the speed of the ego agent was higher than this, then it would experience less pressure to model the partner agent, converging to a policy that was independent of partner properties. Conversely, at lower speeds, the ego agent struggled to converge to an effective policy.

### B.4.2 Experiment 3

In Experiment 3, we test if a 'blind' ego agent that only receives local observations of its own square (position, orientation, and held object) can implicitly model the competence of its partner. For the partner policy, we use the ingredient-preparation subnetwork (subtask 1). During training, the ego agent is paired with a distribution of partners that vary in their probability of taking random actions (versus optimal actions); representing different levels of competence. For testing, we introduce six novel partners with fixed randomness levels $\in \{0, 0.05, 0.1, 0.9, 0.95, 1\}$. This allows us to investigate whether the ego agent adapts its behaviour to both highly competent and highly erratic partners. We hypothesise that the ego will collaborate efficiently with skilled partners, while taking on more of the task when paired with less capable ones.

### B.5 Environment layouts

For both training and evaluation, we use five environment layouts with different spatial constraints: *cramped room*, *coord ring*, *fivebyfive*, *cramped_room_v3*, and *cramped_room_v4*. All layouts were chosen on the basis that they incentivise agents to work together rather than independently, and are no larger than 5x5 tiles (to ensure convergence to successful policies). The five layouts are depicted in Figure 8.

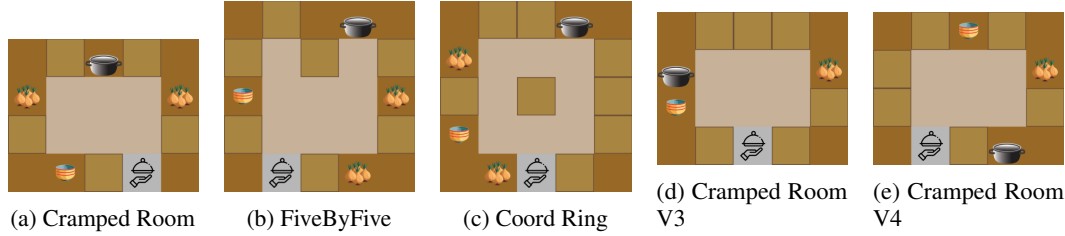

(a) Cramped Room    (b) FiveByFive    (c) Coord Ring    (d) Cramped Room V3    (e) Cramped Room V4

Figure 8: Layouts used in the Experiments

### B.6 Baselines

To isolate the conditions under which partner modelling emerges, we utilise three **baselines** that allow us to disentangle the effects of training diversity, memory, partner information and architecture on collaborative performance. First, to see if memory plays a role, we include a **feedforward network (FFN)** MLP baseline. This tests whether memory is necessary for adapting to diverse partners (poor performance would suggest that memory is important in partner modelling). We next consider a **single partner specialist**, which is a GRU recurrent agent trained with one partner. This baseline probes whether exposure to diverse partners is necessary to develop a generalisable partner model - poor performance in this baseline would outline that training diversity is cruical for emergent modelling. Finally, we used a **non-influential** - an RNN trained over a distribution of partners, but without the ability to influence them. During rollout with the non-influential RNN, the partner mode was switched halfway through training to allow the ego agent to retain memory of the partner's ability in both subtasks.

### B.7 Illustrative videos of agent–partner interactions

To illustrate the experiments and interaction dynamics, we include example videos from the RNN policy trained on a distribution of partners. Although many policies are trained throughout the experiments, these examples are used to highlight the kind of adaptations (and occasional failures to adapt) that emerge under novel test conditions. Videos are included to show the ego agent interacting with two previously unseen partners (using one random seed per layout) in each experiment. In Experiment 1, the agent interacts with a fixed partner throughout the episode — one that is competent

(i.e., speed 1) at ingredient preparation and one at serving. In Experiment 2, the partner switches partway through the episode (at time step 300), from a skilled ingredient preparation partner to a skilled serving partner, or vice versa. During training, partner switches occur at variable times or not at all, so the agent cannot anticipate when or whether a change will occur. In Experiment 3, the blind ego agent is paired with both an unskilled partner and a competent one (at ingredient preparation) - to visualise the large variations in partner ability. These videos are provided in the supplementary materials and demonstrate how the trained policy generalises to new human collaborators.

## B.8  Compute

All experiments were run on A100 GPUs, totalling 462 GPU-hours. Policy training used 225 GPU hours for all three experiments. A total of 37,400 rollouts were simulated, totalling 207 GPU-hours.

## B.9  Code availability

The full code for this paper is available at: `https://github.com/ruaridhmon/emergent_partner_modelling`

