# OpenReview forum: "Partner Modelling Emerges in Recurrent Agents (But Only When It Matters)"
_NeurIPS.cc/2025/Conference — NeurIPS 2025 poster_

### Official Review · Reviewer_ocq2 · 2025-06-02

**Clarity:** 4
**Significance:** 4
**Originality:** 4
**Rating:** 5
**Confidence:** 4

**Summary:**

This paper explores whether and when recurrent agents, trained to perform a collaborative task with another agent, develop representations of their partners "skills". The authors find that, when the recurrent agents are able to interact with their teammate (through assigning their teammate one of two tasks), encoding of the partner's skills emerges in the recurrent agents. The authors further demonstrate that their recurrent agents are able to adapt when their teammate switches skills halfway through the trial. This work provides evidence that artificial models can develop representations relevant for collaborative tasks, even in the absence of any added architectural features.

**Questions:**

Q1 - Why do the authors think the RNN struggled (relative to the baseline models) on the configuration fivebyfive_v1?

**Ethical Concerns:**

["NO or VERY MINOR ethics concerns only"]

**Final Justification:**

I recommend accepting this paper as it is of high quality and interest to the NeurIPS community

**Limitations:**

The authors do a good job addressing their work's limitations. The only exception to this is that they do not note that linear decoding is just one way in which to measure the representation of the partner's skills in their RNN model. Other, more sophisticated strategies of decoding could yield higher accuracy, even for the baseline models.

**Paper Formatting Concerns:**

I did not notice any major formatting issues with this paper.

**Quality:**

3

**Strengths And Weaknesses:**

STRENGTHS:

1. This paper was especially well motivated and written. It was a pleasure to read.

2. The experimental design was, for the most part, very well done and allowed for clear conclusions. I especially appreciated the well chosen controls/baselines that the authors compared their recurrent agents against.

3.  This work tackles a really important question and the results have important implications for how we understand what is needed to design artificial agents that can cooperate effectively. I especially like the way the authors put it - "social intelligence can emerge from specific environmental pressures acting in concert with general mechanisms for learning and memory, rather than necessarily relying on unique architectures". I find it likely that this work will help shape a lot of future research.

4. The future directions noted by the authors are interesting and help frame the impact of the work.

WEAKNESSES:

1. While strong, I think the details on the experiments could be improved in a few specific instances:
    a. The authors mention (in Sec. 5.1) that the recurrent agents were able to generalize to partners that had been unseen in training. However, there is some overlap in the skill range that is sampled during learning ({1, 2, 3, 4, 7, 9}) and the skill range that is sampled during testing ({0, 2, 3, 8, 10}). Do the authors explicitly ensure that the same partner configuration can't emerge in both train and test (e.g., v = (2, 3))? If not, that's fine, but this overlap should be noted.
    b. The online adaptation to a partner that switches is a cool experiment and the way of testing by switching from the partner being better at task 1 and then task 2 at the halfway point makes the analysis clear. But, I was unclear if the recurrent agent was trained on a similar set-up. That is, if it was trained on partners that switched at the halfway point and went from being good at task 1 to task 2. If so, the recurrent agent could learn to switch its policy without actually paying attention to its partner. I think this is not what the authors are doing, but it would be helpful to get more details.

2. While I agree with the authors that their results demonstrate clearly that recurrent agents trained on a diversity of partners outperforms the other baseline models, the single-partner RNN and the no-influence RNN both appear to have information in the recurrent activations that can be used to determine their partner's skills (Fig. 3B - above chance performance for all models). In addition, the authors use a simple linear decoder - there may be more complex decoding schemes in which the activations of the baseline models can lead to even higher accuracy. Therefore, saying that partner modeling only emerges when the RNN can interact with its partner feels a little misleading. I think re-phrasing the results to say that partner modeling emerges more strongly when the RNN can interact with it's partner would be better.

COMMENTS:

1. The UMAP analysis of the hidden activations is interesting, but I think other analysis could help complement this and make the conclusions stronger. For instance, computing the correlation between each hidden unit's average activation and the partner's difference in skill (task 1 - task 2) could lead to the identification of whether the partner modeling occurs at the level of individual units or at the level of the population. The authors could also apply Dynamic Similarity Analysis to identify whether different partner's skills lead to different RNN dynamics.

2. I was intrigued by the fact that the recurrent agent did better than the baseline models on every configuration except fivebyfive_v1 (Fig. 2). Could the authors comment on why this might be?

3. There are a few papers that the authors may be interested in (but, to be clear, are not obligated to cite):
    a. https://elifesciences.org/articles/85694 - This work shows that RL agents can learn to cooperatively hunt (and "assign" roles) without any architecture bias, echoing the authors results.
    b. https://www.nature.com/articles/s41586-018-0102-6 - This work shows that RNN agents trained to perform path integration (i.e., to update where in space the agent is given a sequence of movement directions and speeds) develops computations that are like that in the brain. Leveraging these representations with RL leads to policies that outperform naive RL agents.
    c. https://proceedings.neurips.cc/paper_files/paper/2024/hash/285b06e0dd856f20591b0a5beb954151-Abstract-Conference.html - This work demonstrates that an RNN model that is trained to path integrate only one agent is not optimal for path integrating two agents, illustrating the need for specific computations in multi-agent settings.

---

> ### Author Rebuttal · Authors · 2025-07-30
>
> **Thank you for the detailed and thoughtful feedback.** We’ve clarified the setup, improved the analysis, and incorporated the suggested related work. We appreciate your input and believe it has strengthened the paper.
>
>
> ### Strengths
>
> We appreciate the positive assessment of the motivation, experimental design, and broader implications of our work. We're encouraged that the framing aligns with future research directions.
>
> ### Weaknesses
>
> #### **Weakness 1A: Train/Test Partner Overlap**
>
> Thank you for the great point. While some individual speed values overlapped between training and test, the sets were disjoint in terms of speed pairs (e.g. (2,3)), as all speed pairs used during training were excluded from the test set. This ensured that the ego agent was never paired at test time with a teammate policy it had encountered during training. We apologise for the confusion, and will update the manuscript to make this clearer.
>
> #### **Weakness 1B: Online Adaptation and Training Set-up**
>
> This is also a good point. During evaluation, to simplify the analysis, we always switched the teammate properties in a single direction and at t=300. During training, however, both the timing and the nature of the switch were randomised: the ego agent encountered teammates whose properties switched in both directions (i.e. faster at task 1 → faster at task 2, or vice versa), and t_switch was sampled uniformly between 25% and 75% of the episode length. Additionally, we performed the switch only in 50% of episodes. This was done exactly to mitigate the concern that the model might memorise a fixed timing or pattern, rather than truly adapting to the change online. We will also update the manuscript to clarify this.
>
>
> #### **Weakness 2: Partner Modelling and Decoder Analysis**
>
> Thank you for the feedback; we acknowledge that our wording could be clearer and more specific here. While the baseline models do contain some partner-related information, as evidenced by above-chance decoding performance, the influencing condition shows a substantially stronger emergence of partner modelling. We will update the manuscript to clarify this.
>
>
>
> ### Comments
>
> #### **Comment 1: Additional Analysis**
>
> This is a good point. We chose the UMAP projections as a way to produce clear visual evidence of our findings, and the linear probe analysis for its conceptual simplicity. That being said, we agree that additional analyses would help strengthen our conclusions. The idea of looking at per-unit correlations to better understand at what ‘level’ the representations are being learned is appealing, and has a nice mechanistic interpretability-like flavour. We also thank the reviewer for pointing us to DSA, which we were not previously familiar with. We will make sure to incorporate at least one additional analysis approach into the revised manuscript.
>
> #### **Comment 2: Performance Drop on fivebyfive_v1 Configuration**
>
> This is a good observation - we believe the exception in fivebyfive_v1 highlights that partner modelling is environment dependent. In fivebyfive_v1, spatial coordination is less critical than in cramped_room variants (due to the simpler and more open geography), making feedforward models more competitive. Additionally, these plotted results are from the fully observable setting, where agents can often infer their partner’s behaviour from the current state alone — reducing the benefit of temporal modelling. In contrast, the performance gap to MLPs is larger in the partial and blind settings, where temporal reasoning becomes more important. Another possible factor is that agents are evaluated with novel partners. RNNs may overfit to training partners, whereas MLPs, lacking memory, may generalise better when spatial context is sufficient.
>
> #### **Comment 3: Related Work**
>
> Thank you for sharing these relevant and insightful papers. We’ll incorporate all of the papers into the literature review to make it more comprehensive and to better situate our contributions within the broader context. It was nice to read the papers - they are relevant and great suggestions.

---

> > ### Comment · Reviewer_ocq2 · 2025-08-01
> >
> > I thank the authors for their detailed rebuttal. All my questions/concerns have been addressed. On hearing more about the experimental details, it is clear that the results are well grounded and the interpretations correct. As noted in my review, I enjoyed reading this paper and I look forward to seeing it out (hopefully, in NeurIPS)!

---

### Official Review · Reviewer_we3J · 2025-06-12

**Clarity:** 3
**Significance:** 2
**Originality:** 3
**Rating:** 4
**Confidence:** 3

**Summary:**

This paper investigates whether artificial agents can spontaneously develop internal models of their partners' abilities during cooperative tasks, even without explicit architectural components for modeling others. They find that agents develop structured hidden representations of their partners’ competencies, enabling better task allocation and generalization to unseen partners. Crucially, this modeling emerges only when the agent has some influence over task assignment.

**Questions:**

- See weakness above.

- Why is an RNN used instead of attention? A recent work [1] analyzes collaboration with unseen agents through attention weights. How does this work compare with [1]?

- How does the ego agent collaborate with humans?

- The experiments are conducted in two-agent settings. How does an ego agent collaborate with more agents? Do the current experimental results still hold true? Could the author provide a scalability analysis?

- If the number of patterns is very large (e.g., 20 or 50 patterns), will the ego agent still be able to encode meaningful representations?

- In the partially observable environment, how does the observation radius affect an ego agent’s ability to model partners?

[1] Inverse Attention Agent for Multi-Agent System

**Ethical Concerns:**

["NO or VERY MINOR ethics concerns only"]

**Final Justification:**

The authors have addressed most of my conceptual questions. I still believe that the paper could benefit more from additional experiments on (1) collaboration with humans, (2) scalability with the number of agents, (3) partially observed environments, (4) additional environments, and (5) collaboration with or comparison to attention-based agents.

Nonetheless, the current paper is well-written and easy to follow. It clearly analyzes the current experimental results. Thus, I am recommending a borderline accept.

**Limitations:**

Yes

**Quality:**

3

**Strengths And Weaknesses:**

### Strengths:
- The paper is well-written and easy to follow.
- The use of UMAP visualizations and linear probing to decode partner traits from hidden states provides convincing evidence for internal partner representations.
- The paper includes multiple controlled settings to isolate and validate the conditions under which modeling emerges.

### Weaknesses:

#### Limited Experiments
Many MARL papers conduct experiments on at least two environments and with multiple settings. Do the findings hold true in other environments and under different settings?

---

> ### Author Rebuttal · Authors · 2025-07-30
>
> **Thank you for your thoughtful feedback on our paper.** We appreciate your recognition of the paper’s clarity, the use of analytical tools like UMAP and linear probing, and the careful design of controlled experiments. We’ve addressed all of your points, including obtaining preliminary results in a second environment to strengthen the empirical validation of our claims. We have incorporated, or are incorporating, your questions and suggestions into the revised manuscript, which we believe is stronger as a result and hopefully addresses your concerns.
>
> ### Questions
>
> #### **Q1 Why is an RNN used for attention…**
>
> Thank you for bringing this work to our attention. Long et al.’s paper shares a similar motivation (representation and theory of mind in multi-agent settings), but their approach is different. While we focus on implicit partner modelling, they train agents explicitly to predict/infer each other’s attention. We will update the related work section to clarify this distinction.
>
> Regarding the broader point about RNNs versus attention/transformers, in this work, we chose an RNN because it is a standard architecture for memory-based agents in MARL (Foerster et al., 2016), and offers simple, interpretable internal dynamics. Our aim was to explore whether structured representations emerge without strong architectural inductive biases, and the RNN is well suited to that goal.
>
> While transformers could also be applied here, they typically require larger datasets or pretraining to be effective (Chen et al., 2021), which makes it less suitable for our setting. Additionally, to our knowledge, transformers are mainly used in offline RL (Chen et al., 2021). That said, using transformers to probe agent representations (particularly via attention) would be a promising direction for future work, and we now discuss this explicitly in the revised manuscript.
>
> #### **Q2. How does the ego agent collaborate with humans?**
>
> We do not evaluate collaboration with human partners, but this is an important direction for future work. A key challenge is mutual adaptation: unlike fixed-policy partners, humans adjust in response to the agent, changing the learning dynamics. Understanding how internal representations evolve under such bidirectional influence would be especially valuable. We have updated the manuscript to highlight this as a promising area for future work.
>
> #### **Q3. Scaling to more agents**
>
> This is an interesting direction for future work, and we have updated the manuscript to discuss it more explicitly. We see no reason why the same kind of implicit representation would not scale to more than one teammate, provided the environment imposes sufficient pressure—i.e. all teammates’ behaviour meaningfully affects task success. That said, inference becomes more demanding with additional agents, as more evidence is required to identify their individual properties. The ego agent may adopt useful approximations, such as modelling the average behaviour of its teammates or inferring which teammate is best suited to each subtask.
>
> #### **Q4. If the number of patterns (?) is very large…**
>
> Thanks for the question. In our experiments, the ego agent is exposed to over 50 unique partner pairings, with most variation along the axes of speed and randomness. As partner diversity increases across other behavioural dimensions, we would expect the agent’s internal representations to expand accordingly—encoding not just speed, but also traits like reliability, preferences, or coordination style. Exploring this broader space is an interesting next step. We will update the manuscript to highlight this as an interesting future direction.
>
> #### **Q5. In the partially observable environment…**
>
> In a partially observable environment, the ego agent can't always see its partner and must rely on memory to infer their behaviour. It can still model the partner, but this takes longer, as observable evidence of the partner’s attributes is more sparse. With a radius of 1 (a 3×3 grid centred on the agent), the agent gets some local observations and can adapt fairly quickly. In the blind case, with no visual input, adaptation is slower (but does still occur). The RNN was crucial here, allowing the agent to use past experience to model both the partner and the environment state—for example, it may try to pick up soup, notice no onion arrives, infer the partner is incompetent, and adjust its strategy accordingly.
>
> ### Limitations / weaknesses
>
> #### **Do the findings hold true in other environments and under different settings?**
>
> To address this point and test the robustness of our findings, we have conducted a preliminary experiment in an additional environment; and we will expand upon these results in the revised manuscript.
>
> We agree that demonstrating generalisation beyond Overcooked-AI is a critical next step. Our goal in this paper was to establish the emergence of partner modelling under minimal assumptions, using a rich cooperative domain that allows for clear analysis of agent memory and adaptation. That said, we expect the underlying mechanism — implicit modelling driven by reward and influence — to be broadly applicable. We believe similar dynamics are likely to emerge in other cooperative settings where agents must adapt to diverse partners, especially under partial observability.
>
> To validate this, we conducted preliminary experiments in a second environment: a modified version of CoinGame (from the JaxMARL suite). In our version of the environment, the ego agent must collaborate with a teammate to collect red and blue coins in a small gridworld, and is rewarded based on the total number of coins collected (i.e. fully cooperative reward function). While the ego agent can collect both coin colours, the teammate is only able to collect either red or blue – and, analogously to our overcooked setup, the ego agent has an additional action allowing them to control which colour coins the teammate attempts to collect. Consistent with our results on overcooked, we find the following:
>
> - RNN agent with influence + trained alongside both teammate types outperforms RNN without influence, RNN trained only alongside a single partner, and MLP (in terms of total coins collected) (Table 1)
> - In evaluation episodes with the influence RNN agent, the teammate spends close to 80% of their time pursuing the “correct” coin colour (relative to a ~50% baseline) (Table 1)
> - Teammate types are more (linearly) recoverable from the hidden states of RNN agents with influence than those without (Table 2)
>
> **Table 1** (mean/std taken over 5 training seeds and 1000 eval eps):
>
> | Agent              | Mean coins collected (std) | Mean teammate time on best task (std) |
> |--------------------|----------------------------|----------------------------------------|
> | MLP                | 17.2 (0.483)               | 0.529 (0.0139)                         |
> | RNN no-influence   | 15.5 (0.216)               | 0.485 (0.0000)                         |
> | RNN single-partner | 16.3 (0.515)               | 0.485 (0.0036)                         |
> | RNN full           | 25.7 (1.97)                | 0.793 (0.0471)                         |
>
> **Table 2** (mean/std taken over 5 training seeds and 1000 eval eps):
>
> | Agent              | Mean linear probe accuracy (std) |
> |--------------------|----------------------------------|
> | RNN no-influence   | 0.703 (0.0451)                   |
> | RNN single-partner | 0.639 (0.0215)                   |
> | RNN full           | 0.874 (0.0594)                   |
>
> These results suggest that the phenomenon is not limited to Overcooked, but instead reflects more general properties of cooperative RL under the right environment pressures.
>
> ### References
>
> Chen, L., Lu, K., Rajeswaran, A., Lee, K., Grover, A., Laskin, M., & Abbeel, P. (2021). Decision Transformer: Reinforcement Learning via Sequence Modeling. Advances in Neural Information Processing Systems, 34, 15084–15097.
>
> Foerster, J. N., Assael, Y. M., de Freitas, N., & Whiteson, S. (2016). Learning to Communicate with Deep Multi-Agent Reinforcement Learning. Advances in Neural Information Processing Systems, 29.

---

> > ### Comment · Reviewer_we3J · 2025-08-04
> >
> > I thank the authors for their detailed response. I will raise my score.

---

### Official Review · Reviewer_FxZn · 2025-06-17

**Clarity:** 3
**Significance:** 2
**Originality:** 3
**Rating:** 4
**Confidence:** 5

**Summary:**

The authors explore the learned representations of RNN-based RL agents using population-based training in the game Overcooked. They explore whether these agents are implicitly learning to model partner behavior in response to different environment pressures. To do so, they rely on several factors, from the time an agent takes to accomplish a subtask, to dimensionality reduction techniques on its hidden states, to variations in reward and throughput between different training techniques. They find that agents are capable of generalizing to meaningfully different partners without hidden state resets, that agents following egocentric observations still are capable of modeling partner behavior, and that the hidden state representations for these PBT agents cluster agent behavior much better than baselines.

**Questions:**

- Your findings are demonstrated exclusively in Overcooked-AI. Can you provide evidence or discussion regarding how the observed emergence of partner modeling might generalize to other cooperative domains, especially those with more complex or less structured task decompositions (e.g., sequential social dilemmas or zero-sum games implemented in JAX frameworks)? Are there plans or preliminary results for such extensions?
- The paper focuses on relatively simple partner representations (task speeds for two subtasks). How would your approach handle more complex or overlapping subtasks, or scenarios where partner traits are less easily parameterized? Have you considered or attempted experiments in such settings, and do you expect reward-driven implicit modeling to scale in fidelity compared to explicit partner modeling?
- The related work section is somewhat limited, missing literature on partner modeling, zero-shot coordination, representation learning, and especially meta-learning. Given that RNNs trained with diverse partners are effectively performing memory-based meta-learning, can you expand the literature review to better situate your contribution within these fields? Are there key works from cognitive science, anthropology, economics, or RL that would help motivate your hypothesis that prediction is central to intelligence?
- Have you compared your RNN-based implicit partner modeling to models that use explicit partner modeling modules? For example, how do value estimates, adaptation speed, or representation fidelity differ between your approach and architectures with explicit partner modeling? Is the mapping from hidden state to partner speed similar to that from explicit models?
- Your discussion could be enriched by explicitly addressing the relationship between your findings and large language models (LLMs), particularly those trained via RLHF on human preferences. Do you see parallels between the emergence of partner modeling in your agents and the behaviors of LLMs in collaborative or alignment settings?

**Ethical Concerns:**

["NO or VERY MINOR ethics concerns only"]

**Final Justification:**

I will keep a score of 4 and believe this paper should be accepted into the conference. It is a valuable insight which has implications for how we think about the design of other multi-agent systems such as LLMs, which might benefit from additional conversational pressures that require better anticipating others' behaviors. The paper could benefit from additional environments and baselines, but these reasons are not sufficient to justify rejection.

**Limitations:**

yes

**Quality:**

3

**Strengths And Weaknesses:**

# Strengths
- Creative experiments. The pressures introduced such as partners completing tasks at different speeds and agents being blind but still following a cooperative reward seem like very clear cases where it might be optimal to predict a partner’s behaviors, and they are quite distinct from much prior work in Overcooked
- The paper uses multiple metrics (reward, adaptation curves, linear probe accuracy, UMAP projections) and appropriate baselines (MLP, single-partner RNN, non-influential RNN) to support its claims
- Results include confidence intervals and are evaluated across multiple seeds and layouts, enhancing reliability
- The paper promises detailed experimental details, code, and data in the supplementary material, supporting reproducibility
- The demonstration that "blind" agents can still develop partner models based only on egocentric signals and reward is a significant and surprising result, as well as the results on reset-free online partner adaptation, which is a much more natural form of cooperation
# Weaknesses
- All experiments are conducted in Overcooked-AI. While this is a rich environment, the generality of the findings to other cooperative domains or real-world settings is not demonstrated, and it would be particularly interesting to apply this evaluation to sequential social dilemmas (https://arxiv.org/html/2503.14576v1) or zero-sum games in Jax
- The partner modeling representations are fairly simple (task speeds) and presuppose there being two subtasks. However, in more complicated settings with overlapping subtasks, it would be interesting to see whether reward alone leads to predictive representations that have the same fidelity as explicit partner modeling
- The related work and overall references to prior work feels rather light, there is a large body of work on partner modeling, zero-shot coordination, and representation learning in RL. While some of the key papers are touched on, it might serve the authors better to motivate their hypothesis more in the introduction by drawing from a wide range of literature in cog sci, anthropology, economics, and RL which suggest prediction may be at core of intelligence and that it’s an optimal strategy. Relatedly, the meta-learning literature isn’t referenced, but it could really enhance the literature review, especially since RNNs training to collaborate with many partners are effectively doing memory-based meta-learning.
- In the discussion of impact, it might be nice to reference the relationship between these findings and LLMs. LLMs trained with RLHF on human preferences are a very similar setup to the agents in this paper, so this work may offer insight into why it works well or what LLMs may be doing under the hood.
- The specific methodology used is incremental, albeit the application case is novel. Studying representations learned by agents in social settings is a very interesting use case, but there are not particularly novel techniques used to do that, even though the experiments are creative relative to what the Overcooked literature has historically featured.
- No comparison to models using explicit partner modeling. It would’ve been nice to see how the value estimates differ between RNNs using explicit partner modeling and not using explicit partner modeling, and if the mapping from the hidden state to partner speed is similar as the ones from partner modeling
- Nitpick: Doing this study with an RNN is fine, but with a transformer the authors could have done more than just predict speed, they also may have been able to track attention to different grid cells as a way of seeing if the memory-based agent is tracking where the other agent is or may be going implicitly
- Nitpick: Figure 1 is very large, and this may be taking space that would be better used to make the other figures harder to read. Figure 2 for instance has a lot of information and the labels aren’t too legible, and 2a may be easier to follow as a bar graph colored by model type and the x ticks being the layout names. Reducing figure size but increasing text size and/or breaking the figures up should make for an easier read and let the authors include more results.

---

> ### Author Rebuttal · Authors · 2025-07-30
>
> **Thank you for the thoughtful and constructive feedback.** We appreciate the recognition of our experimental design, evaluation, and key findings on implicit partner modelling. We have addressed all of the points raised, including conducting preliminary experiments in a second environment, and believe the paper is stronger as a result.
>
> ### Weaknesses
>
> #### **1. Generalisation beyond Overcooked-AI**
>
> We agree that demonstrating generalisation beyond Overcooked-AI is important. Our goal was to show that partner modelling can emerge under minimal assumptions in a rich cooperative domain. We expect the same reward- and influence-driven dynamics to arise in other settings where agents adapt to diverse partners under partial observability.
>
> In response to this point, we conducted preliminary experiments in a second environment: a modified version of CoinGame (from the JaxMARL suite). In our version of the environment, the ego agent collaborates with a teammate to collect red and blue coins in a small gridworld, and is rewarded based on the total number of coins collected (i.e. fully cooperative reward function). While the ego agent can collect both coin colours, the teammate is only able to collect either red or blue – and, analogously to our Overcooked setup, the ego agent has an additional action allowing them to control which colour coins the teammate attempts to collect. Consistent with our results on Overcooked, we find the following:
>
> - RNN agent with influence + trained alongside both teammate types outperforms RNN without influence, RNN trained only alongside a single partner, and MLP (in terms of total coins collected) (Table 1)
> - In evaluation episodes with the influence RNN agent, the teammate spends close to 80% of their time pursuing the “correct” coin colour (relative to a ~50% baseline) (Table 1)
> - Teammate types are more (linearly) recoverable from the hidden states of RNN agents with influence than those without (Table 2)
>
> **Table 1** (mean/std taken over 5 training seeds and 1000 eval eps):
>
> | Agent              | Mean coins collected (std) | Mean teammate time on best task (std) |
> |--------------------|----------------------------|----------------------------------------|
> | MLP                | 17.2 (0.483)               | 0.529 (0.0139)                         |
> | RNN no-influence   | 15.5 (0.216)               | 0.485 (0.0000)                         |
> | RNN single-partner | 16.3 (0.515)               | 0.485 (0.0036)                         |
> | RNN full           | 25.7 (1.97)                | 0.793 (0.0471)                         |
>
> **Table 2** (mean/std taken over 5 training seeds and 1000 eval eps):
>
> | Agent              | Mean linear probe accuracy (std) |
> |--------------------|----------------------------------|
> | RNN no-influence   | 0.703 (0.0451)                   |
> | RNN single-partner | 0.639 (0.0215)                   |
> | RNN full           | 0.874 (0.0594)                   |
>
> These results suggest that the phenomenon is not limited to Overcooked, but instead reflects more general properties of cooperative RL under the right environmental pressures. We will expand upon these results in the revised manuscript.
>
> #### **2. Handling more complex or overlapping subtasks**
>
> Our approach can scale to more complex or overlapping subtasks, but this depends on two key factors: (i) whether the environment applies sufficient pressure to represent nuanced or entangled partner traits (rather than allowing simpler heuristics), and (ii) how observable those traits are in the teammate’s behaviour. If partner traits are behaviourally identifiable and reward-relevant, we expect implicit modelling to scale accordingly. We have clarified this in the revised manuscript.
>
>
> #### **3. Related work**
>
> We appreciate this point and will expand the related work section to better situate our contribution within partner modelling, meta-learning, representation learning, and cognitive science. To support the view that prediction is central to intelligence, we are incorporating several key references, including:
>
> - **Clark (2013)** – *Whatever next? Predictive brains, situated agents, and the future of cognitive science*.
>   *Behavioral and Brain Sciences, 36(3), 181–204.*
> Argues that cognition and action are driven by hierarchical prediction.
>
> - **Byrne & Whiten (1988)** – *Machiavellian Intelligence: Social Expertise and the Evolution of Intellect in Monkeys, Apes, and Humans*.
>   *Oxford University Press.*
> Suggests human intelligence evolved for social prediction, motivating partner modelling.
>
> - **Harsanyi (1967)** – *Games with Incomplete Information Played by “Bayesian” Players*.
>   *Management Science.*
> Formalises agents as reasoning over others’ hidden traits.
>
> - **Wang et al. (2016)** – *Learning to Reinforcement Learn.*
>     *Proceedings of the 38th Annual Conference of the Cognitive Science Society (CogSci), 1315–1320.*
>     Demonstrates that recurrent policies can learn internal learning algorithms, enabling fast adaptation to novel tasks—supporting our use of recurrence for implicit partner modelling.
>
> - **Grosz & Kraus (1996)** – *Collaborative Plans for Complex Group Action*.
>   Highlights that effective collaboration requires commitment to shared goals, even under decentralised or partial knowledge.
>
>
> We are happy to incorporate any additional suggestions for relevant work.
>
> #### **4. Connection to LLMs and RLHF**
>
> We thank the reviewer for raising this connection and have added a discussion to the manuscript. Our findings suggest a parallel between emergent partner modelling in recurrent agents and the internal representations learned by large language models (LLMs) trained via reinforcement learning from human feedback (RLHF). In both cases, models receive only a scalar reward signal (team reward in our case, human preference in RLHF) and nonetheless develop internal representations that support effective interaction. Notably, this modelling arises without explicit objectives for partner inference or prediction, indicating that social reasoning can emerge as a byproduct of reward optimisation under the right pressures.
>
> A particularly relevant point of contact lies in the role of influence. We find that partner modelling emerges more robustly when agents can influence partner behaviour—for example, by assigning tasks. This mirrors findings in LLMs, where richer user modelling appears in interactive, multi-turn settings, where the model’s outputs shape future feedback (Ouyang et al., 2022;). Together, these results point to a general principle: when a learner’s reward depends on the actions of others and the learner can influence those actions, internal models of others are likely to emerge—even in the absence of explicit modelling objectives or architectural biases.
>
>
> #### **5. Methodological novelty**
>
> We appreciate this observation — we agree that both the analysis methods and agent architectures are standard, and we chose them intentionally. Our goal was to demonstrate that structured, partner-specific representations can emerge robustly — not as an artefact of architectural choices or analysis techniques, but as a consequence of the collaborative training setting. That said, we agree that exploring richer architectures or novel analysis methods would be a valuable direction for future work. We will add a note on this in the revision.
>
> #### **6. Explicit vs implicit partner modelling**
>
> We did not compare against explicit architectures, as our aim was to test if implicit modelling emerges in standard agents without scaffolding. We would expect explicit approaches to perform better on the specific traits they target, but the strength of the implicit RNN approach lies in its flexibility: it can represent any partner attribute that supports reward maximisation. A direct comparison — in terms of adaptation speed, value estimates, and generalisation — would be a valuable direction for future work. We will add a note on this in the revision.
>
>
> #### **7. Transformer architecture**
>
> Thank you for this point—it's a good suggestion and we agree it opens up a very interesting direction for future work. Transformers could support additional analyses, such as examining attention over grid cells to investigate whether the agent implicitly tracks the other’s position or trajectory.
>
> In this work, we chose an RNN because it is a standard architecture for memory-based agents in MARL (Foerster et al., 2016), and offers simple, interpretable internal dynamics. Our aim was to explore whether structured representations emerge without strong architectural inductive biases, and the RNN is well suited to that goal.
> While transformers could also be applied here, they typically require larger datasets or pretraining to be effective (Chen et al., 2021), which makes it less suitable for our setting. Additionally, to our knowledge, transformers are mainly used in offline RL (Chen et al., 2021). That said, using transformers to probe agent representations (particularly via attention) would be a promising direction for future work, and we note this in the revised paper.
>
> #### **8. Formatting**
>
> Thank you for this point - we have updated the formatting of the paper accordingly to make the figures easier to read and to help include more results.
>
>
> ### References
>
> Chen, L., Lu, K., Rajeswaran, A., Lee, K., Grover, A., Laskin, M., & Abbeel, P. (2021). Decision Transformer: Reinforcement Learning via Sequence Modeling. Advances in Neural Information Processing Systems, 34, 15084–15097.
>
> Ouyang, L., Wu, J., Jiang, X., Almeida, D., Wainwright, C., Mishkin, P., ... & Christiano, P. (2022). Training language models to follow instructions with human feedback. Advances in Neural Information Processing Systems, 35, 27730–27744.
>
> Foerster, J. N., Assael, Y. M., de Freitas, N., & Whiteson, S. (2016). Learning to Communicate with Deep Multi-Agent Reinforcement Learning. Advances in Neural Information Processing Systems, 29.

---

> > ### Comment · Reviewer_FxZn · 2025-08-02
> >
> > Thanks for the detailed rebuttal! I really appreciate the additional experiments and analyses, as well as the new references. I think expanding the findings to a new environment does help support the overall finding, and agree with the point about RNNs being capable of potentially modeling any necessary latent variable flexibly, even if this will be underperform explicit modeling of a given feature. I will maintain my current score of 4-weak accept, since the arguments on 2 and 6, while plausible, could use some more empirical support or theoretical justification (i.e. point 2 likely requires additional formalisms and experimental extensions to say what constitutes sufficient pressure and observability). I understand this is likely not possible within the scope of this discussion period, but believe this paper still stands as a useful insight where those findings can be built upon in future work!

---

### Official Review · Reviewer_xMnq · 2025-06-27

**Clarity:** 4
**Significance:** 3
**Originality:** 4
**Rating:** 5
**Confidence:** 4

**Summary:**

This work investigates the factors behind partner modelling in RL agents. The authors use the cooperative Overcooked-AI environment, where they train different variants of agents together with a partner agent whose policy is pre-trained. The experiments vary in the perception of the agent-in-training ("blind agent"), the abilities of the agents (ability to direct the partner agent, partner being slow in one task, slow in both tasks, or exhibiting random behaviour) and in the permanence of the pairing (the fixed-policy partner agent may get swapped during an episode). This variety of experiments shows that the main factor of the emergence of partner modelling is being able to direct the partner to perform different tasks, and that even a "blind" agent will still learn to model some aspects of their game partner.

**Questions:**

- Please elaborate on the agent architecture design.
  - In the appendix, an FC layer is mentioned. Is there just one input FC layer? Or is there also an output FC layer that converts the GRU output into actions? Are there any other pre-processing or post-processing steps?
- Figure 2A: Why does the MLP perform better in the five-by-five environment? It seems to also be quite close in the coord_ring environment.
  - This may indicate that partner modelling may also be environment-dependent.
- Figure 3A: Was there any further investigation of the example No-influence RNN for the cramped room v4? There seems to be a clear separation, but not on task speed.
- Linear probing: How fast does the ego agent re-encode the partner’s task speed? I think this would be interesting to see, especially in the context of Figure 4A, to see if this re-encoding delay has an impact on the lower throughput even after 300 epochs.
- Additionally, why does the throughput never recover after the partner switch?
  - If the ego agent was trained with agents who can be both slow and fast with different tasks, shouldn't it also be able to switch policy to the correct task speed? This is especially true considering that the UMAP embeddings  in Figure 4C seem to show that this is the case, yet the throughput is not increased.
- Appendix A.3.2 — You mention "We hypothesise that the ego will collaborate efficiently with skilled partners, while taking
on more of the task when paired with less capable ones." — was this hypothesis tested?

**Ethical Concerns:**

["NO or VERY MINOR ethics concerns only"]

**Final Justification:**

Based on the discussions and rebuttals I continue to recommend acceptance. Most formatting and clarity issues were/will be resolved in the camera-ready version, and I still think, as I said in my comments and review, that this paper will be a good contribution to NeurIPS.

**Limitations:**

The main limitations I can see, which are not mentioned in the papers, are:

- Only two agents in the environment — It is unknown if the ego agent would build more complex internal representations of multiple agents, or converge to a different strategy without partner modelling.
- Fixed policy of the partner agent — Would two learner agents be able to converge to a similar strategy?
- Allowing the ego agent to control the other agent — This to me comes close (but not quite) to the other agent just becoming an extension of the ego agent, with the ego agent just learning to control this different "module" of its own architecture.

**Quality:**

4

**Strengths And Weaknesses:**

### Originality

The experimental setup and the ideas in this work are novel. While it may be simplistic, I think this is a good step in partner modelling. The closest work I am aware of is Jaques et al., [1], where similarly to this work, influence on other agents is examined as a factor in agent performance.

### Quality

The quality of this work is high. All claims are based on performed analyses, and where appropriate, statistical significance testing is also performed.

### Clarity

The clarity is also high; however, I had a few comments as to what could be improved:

- Figure 1 — The speedometers might be a bit confusing — I initially thought they were showing time taken to perform a task, inferring the opposite relationship (i.e., the arrow being on the left is faster, or it takes less time to perform a task).
- Line 135: JAX should be cited.
- Line 221: alongisde $ \to $ alongside
- Line 209 (and elsewhere): wrt $ \to $ w.r.t., or spelled out.
- Figure 3b — there is a gap in the confidence intervals for the multi-partner RNN line plot.
- Linear probes — what are their dimensions? Is it just one layer of input size GRU h_dim $ \to $ output size 1, with the 1 being the speed prediction?

### Significance

The work has a good amount of significance for the field of MARL. The insights of requiring influence over another agent for a better development of ToM are an interesting datapoint, and would be a good starting point to see if this translates to more than two agents, and to more complex environments.

Additionally, the authors provide all the code, including Docker files for reproducibility, so building upon this work should be relatively straightforward.

### References

[1] Jaques et al., 2019. Social Influence as Intrinsic Motivation for Multi-Agent Deep Reinforcement Learning. _Proceedings of the 36th International Conference on Machine Learning_, in _Proceedings of Machine Learning Research_ 97:3040-3049, https://proceedings.mlr.press/v97/jaques19a.html.

---

> ### Author Rebuttal · Authors · 2025-07-30
>
> **Thank you for your thoughtful and detailed engagement with the paper.**  We appreciate your recognition of the paper’s originality, clarity, technical quality, and broader significance. We're pleased the experimental design and partner modelling results were well received. Below, we respond point by point and outline revisions we have made or are making to clarify and expand on the points you raised, which we believe further strengthen the paper.
>
>
> ### Questions
>
> #### **Q1. Please elaborate on the agent architecture design**
> Thank you for the question - we have updated the appendix to clarify this. The architecture includes both input and output FC layers: after the CNN processes the observation, an FC layer maps the embedding to the GRU’s hidden size. The GRU output is passed to separate two-layer MLPs for the actor and critic, producing action logits and a scalar value, respectively. Observations are pre-processed to give each agent a local, self-centred view, and the CNN output is normalised before it enters the GRU.
>
> #### **Q2. Figure 2A: Why does the MLP perform better in the five-by-five environment?**
>
> This is a good point, and we agree that it indicates that partner modelling is environment dependent. We have updated the manuscript to include an acknowledgement and brief discussion of this idea.
>
> In the five-by-five layout, spatial coordination is less critical than in cramped_room variants (due to the simpler and more open geography), making feedforward models more competitive. Additionally, these plotted results are from the fully observable setting, where agents can often infer their partner’s behaviour from the current state alone — reducing the benefit of temporal modelling. In contrast, the performance gap is larger in the partial and blind settings, where temporal reasoning becomes more important.
> Another possible factor is that agents are evaluated with novel partners. RNNs may overfit to training partners, whereas MLPs, lacking memory, may generalise better when spatial context is sufficient.
>
> #### **Q3. Figure 3A: Was there any further investigation of the no-influence RNN in cramped_room_v4?**
>
> Yes we did investigate this but we were not able to find any explanation for the apparent structure of the no-influence UMAP plot for cramped_room_v4. We suspect that it is likely an artifact of the UMAP procedure (a phenomenon which is not uncommon) – as evidence for this, we created corresponding plots using PCA and t-SNE in place of UMAP, and did not see a similar grouping appear (while the structure of points in the influence case was preserved to at least some degree).
>
> #### **Q4-5. Linear probing and throughput after partner switch**
> Thank you for this helpful point — we are revising the manuscript to clarify why the throughput plateau is lower post-switch. The observed drop in throughput after 300 epochs is actually due to an asymmetry in the speed dynamics between the ego agent and the partner agent when their roles are reversed. In the runs plotted, the partner performs serving soup quickly (1 action per step) and preparing ingredients slowly (1 action every 7 steps). Later, this pattern reverses. The ego agent acts at a constant rate (1 action every 3 steps) for both tasks.
>
> Serving soup involves a longer sequence of actions (e.g. retrieving a bowl, ladling soup, serving) than placing onions in a pot. The system performs better when the faster agent handles the more complex task offsetting its length and improving soup throughput.
>
> The rise in the curve reflects the ego agent adapting to this dynamic—learning to coordinate with a changing partner. Once the throughput plateaus, it indicates the agent has successfully adapted to the partner’s new behaviour.
>
> #### **Q6. Appendix A.3.2 — Was the hypothesis about task allocation tested?**
> Yes, Experiment 3 (the blind agent setting) serves as a partial test of this hypothesis. In this setting, the ego agent was paired with partners of varying competence, and the blind agent had to infer its partner’s ability and decide whether to complete the entire task alone or specialise in a subtask.
>
> ### Limitations
>
> #### **Two agents only**
>
> This is an interesting direction for future work, and we have updated the manuscript to discuss it more explicitly. There is no reason to believe that the same kind of implicit representation would not scale to more than one teammate, provided the environment imposes sufficient pressure—i.e. all teammates’ behaviour is relevant to task completion. That said, inference becomes more demanding with additional agents, as more evidence is required to accurately identify their individual properties.  The ego agent may rely on useful shortcuts, such as modelling the average behaviour of its teammates or inferring which teammate is best suited to each subtask.
>
> #### **Two learner agents**
>
> Great question — we expect that two learner agents could converge to similar strategies through mutual adaptation, though likely via different dynamics. Using a fixed-policy partner lets us focus on the specific question of how the ego agent models and adapts to its teammate. This is also related to the next question about the level of ego agent control/influence over their teammate; if both agents are learning, then it wouldn’t make sense to give one agent control over the other’s subtask, and so we would require a more nuanced mechanism for determining subtask allocation. We have added a brief discussion of this point in the revised manuscript as a direction for future work.
>
> #### **Direct control**
>
> This is a fair point. We used an extreme form of ‘influence’ to give us the clearest/simplest test of whether partner modelling emerges driven by environmental pressure (i.e. there is a very strong link between ability to represent teammate properties and overall reward). A more realistic setting might involve some form of communication, where the partner learns to follow (or ignore) high-level instructions (and this could even be bidirectional). We have updated the discussion to highlight this as a direction for future work.
>
> ### Other Comments
>
> Thank you for these helpful suggestions. We have revised Figure 1 to clarify the meaning of the speedometers and avoid potential misinterpretation. We have also cited JAX at first mention, corrected the typo on line 221, and standardised usage of “w.r.t.” throughout. Moreover, we have clarified the linear probe architecture in the text: it consists of a single-layer linear decoder mapping the GRU hidden state to a scalar output (speed prediction).
>
> *Regarding Figure 3B, thank you for flagging this. We're unsure which specific gap in the confidence intervals is being referred to — we’d be grateful for clarification so we can investigate further.*

---

> > ### Comment · Reviewer_xMnq · 2025-08-02
> >
> > Thank you for your detailed response. I am happy with the clarifications and so continue to recommend acceptance. I am looking forward to reading the paper in its revised (and accepted) version!
> >
> > Regarding Figure 3B — The multi-partner RNN (green) line for Partner's 1 task speed loses the shaded confidence internal after around timestep 150. After that timestep, it looks like it is just the mean line.

---

> > > ### Author Response · Authors · 2025-08-03
> > >
> > > We are glad that our clarifications were helpful, and appreciate the kind words. For Figure 3B, we have now confirmed what appears to be a rendering issue with the pdf version of the image (which we missed before); the shaded confidence interval appears as expected in the original figure, and will make sure the issue is fixed in the revised manuscript.

---

### Decision · Program_Chairs · 2025-09-17

**Decision:**

Accept (poster)

**Comment:**

This paper asks whether agents need explicit partner-modeling modules, or whether such representations can emerge spontaneously under the right social pressures. Using Overcooked-AI, the authors train recurrent RL agents with different partners and show that hidden states encode partner skill levels, which they suggest enables rapid adaptation and generalization. Results support the claim that partner modelling emerges most clearly when agents have influence over task allocation. The authors also extend the analysis with preliminary experiments in CoinGame, suggesting the findings are not limited to one domain.

Reviewers were broadly positive. They praised the clarity of the experimental setup, the strong empirical evidence, and the use of multiple probes (UMAP, linear decoding, adaptation curves) to support the claims. Several highlighted the work’s potential to inform research on social intelligence in artificial agents, noting parallels to human theory of mind and even LLM alignment.

Concerns were mainly on scope and depth: all main experiments are in Overcooked, the representations are fairly simple (mostly task speed), and the literature review could be broader. Some reviewers also suggested comparing against explicit partner-modelling architectures or trying transformers for richer analysis, but agreed that the simplicity of the current approach strengthens the central claim.

After rebuttal, the authors added clarifications and preliminary results in CoinGame, addressed questions on training/test splits, online adaptation, and expanded their discussion of related work and future directions. Reviewers were satisfied and in some cases raised their scores. The contribution is limited in scope, but solid, and merits acceptance as a poster.